# The Impact of Lateral Ventricular Opening in the Resection of Newly Diagnosed High-Grade Gliomas: A Single Center Experience

**DOI:** 10.3390/cancers16081574

**Published:** 2024-04-19

**Authors:** Fabio Cofano, Andrea Bianconi, Raffaele De Marco, Elena Consoli, Pietro Zeppa, Francesco Bruno, Alessia Pellerino, Flavio Panico, Luca Francesco Salvati, Francesca Rizzo, Alberto Morello, Roberta Rudà, Giovanni Morana, Antonio Melcarne, Diego Garbossa

**Affiliations:** 1Department of Neuroscience “Rita Levi Montalcini”, University of Turin, 10124 Turin, Italyelena.consoli@edu.unito.it (E.C.); alessia.pellerino@unito.it (A.P.); flavio.panico@unito.it (F.P.);; 2Neurosurgery Unit, “Città della Salute e della Scienza” University Hospital, 10124 Turin, Italy; 3Division of Neuro-Oncology, “Città della Salute e della Scienza” University Hospital, 10124 Turin, Italy; 4Division of Neurosurgery, Santa Corona Hospital, 17027 Pietra Ligure, Italy; diksalvati@gmail.com; 5Division of Neuroradiology, Department of Diagnostic Imaging and Radiotherapy, “Città della Salute e della Scienza” University Hospital, University of Turin, 10124 Turin, Italy

**Keywords:** glioma, glioblastoma, lateral ventricle, ventricle opening, leptomeningeal dissemination

## Abstract

**Simple Summary:**

This retrospective study investigates the validity of lateral ventricle opening during high-grade glioma (HGG) surgery, contributing to the literature by addressing its relationship with postoperative complications in a large cohort of newly diagnosed patients with HGG. The approach of dividing patients into groups with and without lateral ventricle opening enhances the study’s internal validity, allowing for a direct comparison between the two conditions. The analysis of complications, including subependymal dissemination and multifocal progression, contributes to understanding associated side effects. Furthermore, the conclusion that lateral ventricle opening does not significantly increase the risks of dissemination, hydrocephalus or cerebrospinal fluid leakage suggests practical implications for surgery.

**Abstract:**

Given the importance of maximizing resection for prognosis in patients with HGG and the potential risks associated with ventricle opening, this study aimed to assess the actual increase in post-surgical complications related to lateral ventricle opening and its influence on OS and PFS. A retrospective study was conducted on newly diagnosed HGG, dividing the patients into two groups according to whether the lateral ventricle was opened (69 patients) or not opened (311 patients). PFS, OS, subependymal dissemination, distant parenchymal recurrences, the development of hydrocephalus and CSF leak were considered outcome measures. A cohort of 380 patients (154 females (40.5%) and 226 males (59.5%)) was involved in the study (median age 61 years). The PFS averaged 10.9 months (±13.3 SD), and OS averaged 16.6 months (± 16.3 SD). Among complications, subependymal dissemination was registered in 15 cases (3.9%), multifocal and multicentric progression in 56 cases (14.7%), leptomeningeal dissemination in 12 (3.2%) and hydrocephalus in 8 (2.1%). These occurrences could not be clearly justified by ventricular opening. The act of opening the lateral ventricles itself does not carry an elevated risk of dissemination, hydrocephalus or cerebrospinal fluid (CSF) leak. Therefore, if necessary, it should be pursued to achieve radical removal of the disease.

## 1. Introduction

High-grade gliomas (HGG), particularly glioblastoma (GBM), are the most prevalent malignant primary brain tumors in adults [1]. Despite advancements in therapeutic options and management, the survival rate for patients with glioblastoma hovers around 15–18 months, with a 5-year overall survival rate of approximately 5% [2,3,4].

The established standard of care for patients with GBM includes surgical intervention, aiming for gross total resection (GTR), followed by chemo-radiotherapy treatment [5], while numerous experimental protocols involving immunotherapy are gaining momentum [6]. Surgery serves as the initial step in GBM management, and the extent of resection (EOR) is the sole factor within neurosurgeons’ control to enhance prognosis [7,8]. While the definition may vary, achieving GTR could significantly enhance both progression-free survival (PFS) and overall survival (OS) for patients. Consequently, the resulting performance status plays a crucial role in subsequent treatment decisions [9,10,11].

Despite notable advancements in surgical techniques and intraoperative technology, such as Fluorescence Guided Surgery (FGS), intraoperative Magnetic Resonance Imaging (iMRI) and intraoperative Ultrasounds (iUS, the prognosis for GBM remains grim [12,13,14].

This occurs due to the infiltrative nature of the tumor, leading to a majority of relapses at the tumor’s edges [15]. Detection beyond the contrast-enhancing nodule is possible via FLAIR MRI sequences or biologically active regions in PET scans [16]. 

In particular, the integration of fluorescence in glioma surgery enhanced both radiological and clinical outcomes concerning the extent of resection by enabling the visualization of not only the enhancing nodule but also the infiltrating tumor beyond the MR-enhancing lesion [12]. Therefore, considering that the extent of resection is correlated with survival, all these systems have led to an increase in the extent of resection and, thus, in the supratentorial compartment, encountering the lateral ventricle as an anatomical limit [17,18]. Currently, evidence in the literature is not consistent in suggesting its opening as an anatomical target, nor in suggesting to avoid its opening for a significant increase in complications: higher incidence of postoperative hydrocephalus, leptomeningeal dissemination and possible increased incidence of distant parenchymal recurrence are among negative effects which have been related to this circumstance [19], but on the other hand, the presence of glioma cells inside or next to the subventricular zone (SVZ) could justify a ventricular opening even in the absence of contrast enhancing at MRI [18,20,21,22,23,24]. The potential danger and complications of ventricle opening have to be considered by surgeons. Moreover, with regard to predominantly surgical considerations, ventricular opening may present a challenge concerning the utilization of hemostatic matrices containing thrombin, given the potential risk of hydrocephalus. Additionally, the methods for dural closure following ventricular opening constitute a debated subject, as both direct suturing and the application of dural substitutes, whether suturable or non-suturable, could impact the risk of cerebrospinal fluid fistula. The primary aim of this study is to assess the correlation between ventricular opening during high-grade glioma resection and the above-mentioned postoperative complications, with a secondary focus on evaluating its impact on survival compared to cases without ventricular opening.

## 2. Materials and Methods

This monocentric retrospective study analyzed a series of patients with histologically confirmed diagnoses of high-grade glioma (III–IV WHO 2016–2021) [25,26] who underwent craniotomy in a period between January 2015 and August 2021. The study was conducted in compliance with Good Clinical Practice guidelines and according to the ethical standards of the author’s institution.

The inclusion criteria were chosen based on the primary objectives of the paper, as well as the WHO classifications of 2016 and 2021 and previous studies that discussed this topic [20,23]. All adult patients affected by newly diagnosed high-grade gliomas were included in the study, while low-grade gliomas, infratentorial location and multicentric lesions were excluded. The availability of clinical and radiological records was considered essential for recruitability. Specifically, the following data were collected: sex, age, preoperative Karnofsky Performance Status, presence of neurological deficits and seizure history.

Each patient underwent contrast-enhanced Magnetic Resonance Imaging (MRI) at least two weeks before surgery and within 48 h after surgery. Comparing those images, patients were divided according to EOR: subtotal resection (STR) < 95%, gross total resection (GTR) 95–100% or supratotal resection (SupTR), comprehending area outside the contrast-enhancing nodule.

The patients were divided into two groups: one in which the lateral ventricle had not been opened (control group) and one in which it had been opened (ventricle opening group). Two different modes of ventricular opening were distinguished based on post-operative T2-weighted MRI sequences: wide opening (>2 cm) or fissuring (<2 cm). The opened ventricular portion (i.e., frontal horn, temporal and occipital) was recorded. In case of ventricular opening, the dura mater was closed by direct suture when the native dura was available or with a dural substitute, with or without the adjunct of fibrin glue. Finally, in some cases, a lumbar drain (LD) or an external ventricular drain (EVD) was placed.

Postoperative complications such as hydrocephalus and MRI signs of subependymal and leptomeningeal dissemination and multicentric progression were considered primary endpoints, while how the ventricle opening influenced OS was considered a secondary endpoint of the study. Subependymal dissemination is defined as the presence of contrast-enhancing lesions within the ventricles, while leptomeningeal dissemination involves the spread to the leptomeninges outside the ventricles. Multicentric or multifocal progression has been considered the disease progressing with the appearance of new nodular enhancements on T1ce images, whether contained within or outside the visible signal alteration in FLAIR sequences. The evaluation of these forms of disease progression was conducted via contrast-enhanced brain MRI scans performed as part of the oncological follow-up protocol for patients. The assessments commenced in the month subsequent to the surgical intervention and continued until the potential occurrence of lesions without a predefined time limit for follow-up.

Statistical analysis was conducted using an open-source software built on R language (The jamovi project (2021). jamovi. (Version 2.2) [Computer Software], retrieved from https://www.jamovi.org (accessed on 1 December 2022)). Statistical significance was set at *p* < 0.05. A chi-square test was performed for simple correlation. The influence of ventricular opening on OS was analyzed with Kaplan–Meier curves. A generalized mixed model was used to assess the potential association between ventricular opening and complications; furthermore, contingency tables were reported for complications.

## 3. Results

A cohort of 380 patients (154 females (40.5%) and 226 males (59.5%)) was involved in the study. At the date of surgery, the median age was 61 years (range: 20–84; ±12.6 SD). A mean of 85 (±12.6 SD) was obtained for the Karnofsky performance score (KPS).

A total of 53 grade WHO grade III (13.9%) and 327 WHO grade IV (86.0%) gliomas were collected. Regarding the histological tumor type, 324 were glioblastomas (GBM) (85.3%), 28 astrocytomas (7.4%), 3 gliosarcomas (0.8%), 1 oligoastrocytoma (0.3%), 20 oligodendrogliomas (5.3%), 2 xanthoastrocytomas (0.5%), 1 anaplastic ependymoma (0.3%) and 1 anaplastic ganglioglioma (0.3%). In the big group of GBM, 152 tumors (40%) showed an O(6)-methylguanine-DNA methyltransferase (MGMT) promoter methylation, 46 (12.1%) were IDH1 mutated and 3 (0.8%) IDH2 mutated. The average percentage value of ki-67 was 39.1 (0.2–90; ±22.2 SD).

Overall, 171 tumors were located in the right hemisphere (45%), 197 in the left (51.8%) and 12 along the midline (3.2%). Specifically, most lesions (162) were located precentrally in the frontal lobe (42.6%), 76 in the postcentral area (20%) and 142 in the temporo-insular region (37.4%).

Considering the specific portion of the lateral ventricle next to the tumor location, 34 (9.1%) lesions were adjacent to the occipital horn, 154 (41.1%) to the temporal horn and 124 (33.1%) to the frontal horn, while the walls of the third ventricle were the nearest point in 63 cases (16.8%).

In 99 patients, the contrast-enhanced nodule was contiguous to the ventricle (26.3%); in the remaining patients, the distance varied: <0.5 cm in 45 (11.8%), between 0.5 cm and 1 cm in 51 (13.4%), between 1 and 1.5 cm in 47 (12.4%), between 1 and 2 cm in 43 (11.3%), between 2 and 2.5 cm in 43 (11.3%) and over 2.5 cm in 51 patients (13.4%).

Preoperative MRI showed a meningeal involvement in 13 patients (3.4%), an ependymal involvement in 21 (5.5%) and the corpus callosum was infiltrated in 39 cases (10.3%).

A GTR was obtained in 305 patients out of 380 (80.3%) (26 SupTR (6.8%) and 49 STR (12.9%)). The ventricle was opened 69 times (18.2%): 37 were wide openings (52.9%) and 32 were fissures (47.1%).

A total of 281 (73.9%) patients received chemotherapy with Temozolomide (50 in the ventricle opening group), while 265 (69.7%) patients underwent concomitant postoperative radiotherapy (47 in the ventricle opening group), according to Stupp or Perry protocols.

The dura mater was closed via direct suturing in cases with approximable margins and the absence of neoplastic infiltration. Dural substitutes, whether suturable or non-suturable, were used based on the preferences of the first surgeon in cases of incidental dural laceration during craniotomy or if the aforementioned criteria were not met.

The most frequent type (161/380 (42.4%)) of closure was characterized by the application of a non-suturable dural substitute plus fibrin glue. A direct suture of the dura plus dural substitute and fibrin glue 102 times (26.8%). Direct suturing of the native dura without any adjunct of fibrin glue was performed in 30 patients (7.9%) and by direct suture plus fibrin glue in 43 (11.3%); a suturable dural substitute without fibrin glue was used in 23 cases (6.1%) with the addition of fibrin glue in 21 cases (5.5%)

A lumbar drain was placed preoperatively in those patients in whom it was deemed necessary to achieve brain retraction or deliquoration to facilitate surgical access and removed at the end of the procedure, while the EVD was placed postoperatively in the surgical cavity in cases where the tumor had a large intraventricular component.

An EVD was placed in six patients (1.6%) and an LD in two patients (0.5%).

A Chi-square test (Table 1) was performed considering a few variables, such as the type of tumors or distance from the ventricle, with respect to ventricle opening.

The probability of opening the ventricle was higher in case of a smaller distance from it (*p*-value < 0.01). Furthermore, similarly to other studies, in the presence of an ependymal involvement, the probability of opening the ventricle was significantly higher.

Tumor characteristics (type or grade, *p* = 0.29 and 0.59, respectively), molecular mutations (MGMT methylation, *p* = 0.48; IDH1, *p* = 0.89; IDH2, *p* = 0.49), leptomeningeal and corpus callosum involvement (*p* = 0.79 and 0.20, respectively), Ki-67 (*p* = 0.40) and the placement of a lumbar drain (*p* = 0.50) did not show any relationship with entering the ventricle.

The mean preoperative KPS was similar in the ventricle opening group (84.9 (±12.4 SD)) and in the control group (85.0 (±12.7 SD)). A good functional outcome was preserved in the majority of patients (postoperative KPS mean value of 80.5 (±18.1 SD) without significant difference between ventricle opening group (79.5 (±18.1 SD)) and control group (80.6 (±18.1 SD)).

The PFS averaged 10.9 months (±13.3 SD) in all population, in particular 9.1 months (±16.3 SD) in ventricular opening group and 11.2 months (±13.3 SD) in control group. OS averaged 16.6 months (±16.3 SD) and ranged from 10.9 months (±13.3 SD) in the ventricle opening group to 17.1 months (±16.3 SD) in the control group.

Among complications, subependymal dissemination was registered in 15 cases (3.9%), multifocal and multicentric progression in 56 times (14.7%), leptomeningeal dissemination in 12 (3.2%) and hydrocephalus in 8 (2.1%). These occurrences could not be clearly justified by ventricular opening (Table 2).

A generalized mixed model was used to assess the potential association between ventricular opening and subsequent subependymal dissemination, considering confounding factors such as the type and grade of neoplasm, MGMT methylation status, IDH mutation status, as well as its distance from the ventricle. The same analysis was then repeated using leptomeningeal dissemination, multicentric progression and development of hydrocephalus or cerebrospinal fluid leak as outcomes. No statistically significant association was found between intraoperative ventricular opening and the subsequent development of leptomeningeal or subependymal dissemination, complications or multicentric progression (Table 3).

Similarly, no statistically significant associations were found between the following:-Distance of the enhancing portion of the tumor and dissemination, multicentric progression or development of hydrocephalus;-Site of ventricular opening and dissemination, multicentric progression or development of hydrocephalus;-Type of dural closure and complications.

As expected, however, the association between tumor distance from the ventricle and ventricular opening was strong (*p* < 0.001 even when stratified by tumor type and grade).

A survival analysis (Kaplan–Meier estimator) in patients with ventricular opening versus those without opening at 6, 12 and 24 months showed a slightly worse OS. Specifically, in the control group, survival at 6, 12 and 18 months was 78.2% [73.7–82.9%, 95% CI], 61.3% [55.9–67.2%, 95% CI] and 42.9% [37.2–49.4%, 95% CI], respectively. On the other hand, the percentages of patients who experienced ventricular opening were 68.1% [57.7–80.4%, 95% CI], 37.1% [26.6–51.7%, 95% CI] and 32.8% [22.6–47.6%, 95% CI], respectively. The curves were adjusted for the following variables (age, KPS, WHO grade, MGMT and IDH status and adjuvant therapy), with no differences observed between the two groups.

The risk of death in patients who underwent ventricular opening was 1.31 times higher (0.94–1.82, *p* = 0.109). However, these results were not statistically significant. (Figure 1)

## 4. Discussion

Surgery and extent of resection (EOR) appeared to impact the most on overall (OS) and progression-free survival (PFS) [7,27]. However, it is essential to balance EOR with a good functional outcome, aiming for a maximal safe resection [28,29,30]. In recent times, maximal safe resection has become the standard of care for patients with glioblastoma, implying the most radical removal feasible while preserving the patient’s neurological functionality. Indeed, awake surgery with brain mapping, advanced imaging modality and different types of intraoperative dyes are among the factors that can contribute to increased EOR, being, however, as safe as possible [31,32,33]. This has led to the redefinition of anatomical boundaries, increasing the significance of functional limits. Nevertheless, in the surgical planning of the extent of resection in high-grade gliomas, the anatomical boundaries themselves, such as reaching the lateral ventricle, must be considered. This aspect remains contentious, as the decision to open the ventricle introduces a complex set of considerations due to its potential association with postoperative complications. These complications encompass leptomeningeal dissemination, hydrocephalus, cerebrospinal fluid (CSF) leakage and multifocal multicentric progression, posing challenges that could adversely impact overall survival. Therefore, the debate surrounding ventricular opening reflects the nuanced interplay between achieving optimal resection and carefully managing the risks associated with potential postoperative complications [20,24,34,35,36].

However, this aspect is still controversial. The Subventricular Zone (SVZ), located along the margin of the Caudate Nucleus adjacent to the lateral ventricle, is one of the areas where neurogenesis occurs, as it serves as a source of neural stem cells. The connection between SVZ and glioblastomas is particularly interesting because the SVZ is a site of persistent neurogenesis even in adults, which suggests that genetic alterations in this context may contribute to the formation of such tumors. It has also been hypothesized that the SVZ could represent a potential therapeutic target. Indeed, Matsuda et al. [37] showed that leptomeningeal dissemination was more probable in the case of SVZ involvement than with ventricular opening during surgery. Furthermore, an association between distant and multifocal recurrence and SVZ involvement has already been reported in the literature [38,39], which was sustained by the hypothesis of the similarity of glioma cells in SVZ with neural stem cells. This more aggressive and invasive behavior could be explained by a different cellular environment [40,41]. Jafri et al. [42] showed a worse rate of PFS and OS in the case of SVZ involvement, and more recently, Saito et al. [17] obtained a better rate of OS by increasing the amount of the SVZ during glioma resection in a comparison between wide and small ventricular opening. Furthermore, other studies did not find an association between ventricular opening and the appearance of such complications [13,16]. Since this maneuver does not strictly cause dissemination or hydrocephalus, it may be justified to increase the extent of resection when safe.

Behling et al. [21] demonstrated that ventricular opening, tumor size or proximity to the ventricle and EOR did not emerge as significant risk factors for hydrocephalus. Ventricular opening did not result in a delay in postoperative therapy and did not correlate with neurological morbidity.

The current study contributes to feeding this contrasting neurosurgical scenario, in particular highlighting how no significant differences in complications have been observed between opening and not opening the lateral ventricle, regardless of the method of opening, in a sample size that is far from negligible. Regardless of ventricular opening, the greater malignancy of lesions in the periventricular region is confirmed, as they appear more infiltrative, deeper, and thus burdened with increased morbidity and a lower rate of GTR. However, among the factors limiting the extent of resection, both functional and anatomical, reaching the lateral ventricle should not be included. It can serve as a valuable landmark, especially after losing the accuracy of navigation due to the brain shift phenomenon, without carrying a significant risk of complications. In addition, the method of dural closure may be at the discretion of the surgeon and based on the individual clinical case. Even in the case of ventricular opening, our case analysis does not indicate an increased risk of cerebrospinal fluid fistula with the use of non-suturable dural substitutes or specific dural closure methods. Likewise, there is no empirical evidence indicating an elevated risk of hydrocephalus associated with the application of hemostatic matrices containing thrombin, even in the case of ventricle opening. These considerations substantiate the proposition that ventricular opening should be undertaken if deemed essential for surgical objectives. Importantly, such a maneuver does not necessitate modifications to hemostatic or dural closure methodologies, ensuring the preservation of established surgical procedures without imposing additional time-consuming strategies.

Kaplan–Meier curves showed that survival at 6, 12 and 18 months was higher in patients without ventricle opening. In any case, survival in gliomas is influenced by a multitude of factors (such as location, extent of resection, molecular factors and adjuvant therapy). Ventricular opening falls within this broad spectrum of variables. A possible explanation could be in the location of lesions with extension in the ventricle that may be deeper and more complex tumors; in these cases, it is more difficult to achieve complete resection, and there is a greater risk of postoperative neurological damage, which can impact adjuvant therapies and survival itself. However, due to the absence of statistical significance (*p* = 0.11), the negative effect on survival that has been described after ventricular opening should not be so obvious.

Similarly, a Chi-squared test did not find any significant cause-effect association between ventricular opening and the appearance of hydrocephalus, multifocal or multicentric progression, leptomeningeal and ependymal dissemination.

In an evaluation between complications and distance of CEN from the ventricle, the multifocal/multicentric tumor progression emerged statistically significant (X^2^ = 4.82, *p* = 0.028). As a result, a close tumor–ventricle distance and ependymal involvement needed a more aggressive surgical intervention, impacting the outcome.

Ventricular opening is part of the multitude of choices that the surgeon must make in order to guarantee maximal safe resection. Its proximity to the ventricle a priori correlates with a worse prognosis due to its deep localization and the associated risks described, but its involvement in the surgical procedure does not seem to be significantly associated with the development of post-surgical complications; therefore, if necessary (tumor–ventricle proximity or ependymal involvement), its opening should be evaluated.

## 5. Limitations

A few limitations have to be highlighted: the retrospective and monocentric nature of the analysis makes the results less meaningful. Indeed, a multi-center study could provide more generalizable insights into the impact of ventricular opening on the outcomes of patients with high-grade gliomas. The groups were not randomized. In addition, a CSF sample was not taken after ventricular opening to determine whether glioma cells had spread.

## 6. Conclusions

Opening the lateral ventricle per se is not associated with an increased risk of leptomeningeal dissemination, multicentric progression, hydrocephalus, or CSF leak following surgery. These data support the surgical choice to pursue maximal resection without the fear of opening the ventricle in order to achieve complete removal of the disease.

## Figures and Tables

**Figure 1 cancers-16-01574-f001:**
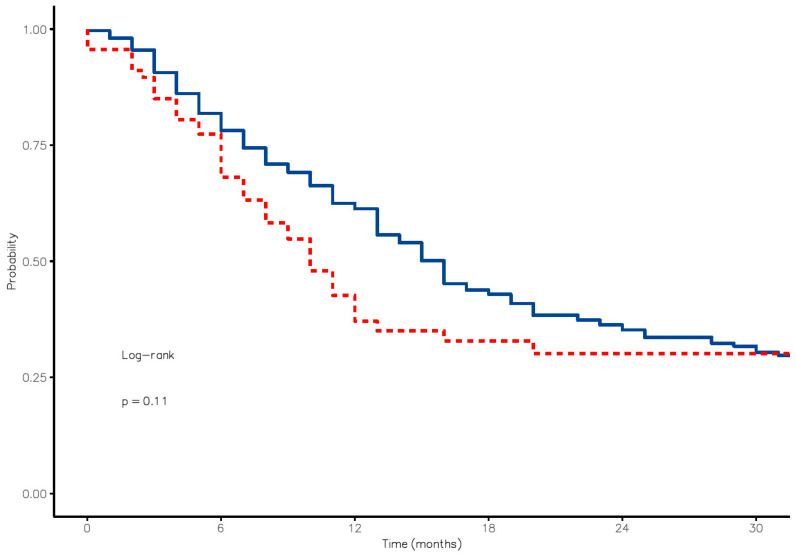
Kaplan–Meier curves distinguishing the OS in patients experiencing ventricle opening (red line) by those where the ventricle was preserved (blue line).

**Table 1 cancers-16-01574-t001:** Association between preoperative and intraoperative characteristics and the chance of ventricular opening.

Variables	No Ventricular Opening	Ventricular Opening	Statistics
**Location**			X^2^ = 9.44 *p* = 0.01
Precentral	144/311	17/69	
Postcentral	58/311	18/69	
Temporoinsular	109/311	33/69	
**Distance of CEN from Ventricle**			X^2^ = 34.0 *p* < 0.01
Contiguous	65/311	34/69	
<0.5 cm	33/311	12/69	
0.5–1 cm	46/311	5/69	
1–1.5 cm	44/311	3/69	
1.5–2 cm	39/311	4/69	
2–2.5 cm	39/311	4/69	
>2.5 cm	45/311	6/69	
**Tumor Type**			X^2^ = 8.47 *p* = 0.29
Glioblastoma	267/311	57/69	
Astrocitoma	25/311	3/69	
Gliosarcoma	2/309	1/69	
Oligoastrocitoma	1/309	0/69	
Oligodendroglioma	15/311	5/69	
Xantoastrocitoma	1/311	1/69	
Anaplastic Ependymoma	1/311	0/69	
Anaplastic Ganglioglioma	0/311	1/69	
**Leptomeningeal involvement**	11/311	2/69	X^2^ = 0.07 *p* = 0.79
**Ependymal involvement**	12/311	9/69	X^2^ = 9.37 *p* < 0.01
**Corpus Callosum involvement**	29/311	10/69	Χ^2^ = 1.64 *p* = 0.20
**Lumbar Drain (LD) positioning**	2/311	0/69	Χ^2^ = 0.45 *p* = 0.50
**External Ventricular Drain (EVD) positioning**	0/311	6/69	Χ^2^ = 27.48 *p* < 0.01
**Extent of Resection (EOR)**			Χ^2^ = 2.53 *p* = 0.28
SupTR	24/311	2/69	
GTR	249/311	55/69	
STR	38/311	11/69	

**Table 2 cancers-16-01574-t002:** Contingency tables of complications.

	No Ventricular Opening	Ventricular Opening	
Hydrocephalus			Total
No	305	67	372
Yes	6	2	8
Total	311	69	380
X^2^	0.257		*p* = 0.612
Leptomeningeal Dissemination			Total
No	302	66	368
Yes	9	3	12
Total	311	69	
X^2^	0.390		*p* = 0.532
Subependymal Dissemination			Total
No	300	65	365
Yes	11	4	15
Total	311	69	380
X^2^	0.761		*p* = 0.383
Multifocal/Multicentric Progression			Total
No	265	59	324
Yes	46	10	56
Total	311	69	380
X^2^	0.004		*p* = 0.950

**Table 3 cancers-16-01574-t003:** Results of the mixed linear model, considering as confounding factors the type and grade of neoplasm, MGMT methylation profile, IDH mutation status and distance from the ventricle.

	Ventricular Opening (*p*-Value)
Hydrocephalus	0.597
Leptomeningeal Dissemination	0.515
Subependymal Dissemination	0.369
Multifocal/Multicentric Progression	0.979

## Data Availability

The data that support the findings of this study are available from the corresponding author, A.B., upon reasonable request.

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
