# Peer review of "The Impact of Lateral Ventricular Opening in the Resection of Newly Diagnosed High-Grade Gliomas: A Single Center Experience"

_cancers, 2024, doi:10.3390/cancers16081574_

Round 1
Reviewer 1 Report (Previous Reviewer 1)
Comments and Suggestions for Authors
In this retrospective study the authors shown that the lateral ventrical opening does not affect the frequency of complications, such as subependymal dissemination and multifocal progression, as well as hydrocephalus, or cerebrospinal fluid leakage. This knowledge is indeed useful for implication from the point of view of increasing the level of radical tumor removement. After the first turn of revision the study may be published as is.
Author Response
We would like to thank the reviewer for their positive comments and for contributing to the improvement of the manuscript.
Reviewer 2 Report (Previous Reviewer 3)
Comments and Suggestions for Authors
I thank the authors for addressing my comments and for their responses. I have read the latest version of their manuscript and I concur with all the changes implemented by the authors. I am sure these will benefit the potential reader. Therefore, I recommend the manuscript to be accepted for publication in its current version.
Author Response
We express our gratitude to the reviewer for their insightful and constructive feedback, which has significantly enhanced the quality of the manuscript
Reviewer 3 Report (New Reviewer)
Comments and Suggestions for Authors
1. The comma is not necessary in the title.
2. In section 2, the statistical significance is usually defined as P<0.05 instead of P≤ 0.05.
3. In section 3, using the median instead of the mean to explain the age distribution is better.
4. It is better to rearrange Table 3. The letter “P” only needs to emerge once in the first row.
5. For the survival analysis, the author should emphasize the effects of Temozolomide administration and radiotherapy individually or in combination. The potential contribution of the chemo- and radio-therapy to OS should be separated from that of the craniotomy.
Author Response
Regarding points 1 to 4, we have made the requested modifications. We removed the period from the title, replaced "P≤ 0.05" with "P<0.05," and changed the presentation of age in the abstract and results sections from mean to median. We have also restricted "p-value" to the first row only in Table 3.
Regarding point 5, we have addressed the issue by clarifying in lines 236-239 that the majority of patients in our study with high-grade gliomas received concurrent chemoradiotherapy protocols if their KPS score was above 70, following either the Stupp regimen or hypofractionated radiotherapy as per Perry's guidelines for elderly patients. We have emphasized the direct impact of therapy on survival. We have chosen not to include a Kaplan-Meier plot adjusted for adjuvant treatment because it was evident that patients receiving such treatment had longer survival, regardless of lateral ventricle opening during surgery. Furthermore, as previously stated in the revision, survival is not the primary outcome of our study; rather, it is the incidence of complications related to lateral ventricle opening. However, we have specified in the results (lines 316-318) that survival analysis was adjusted for the following variables: age, KPS, WHO grade, MGMT, and IDH status, as well as adjuvant therapy.
This manuscript is a resubmission of an earlier submission. The following is a list of the peer review reports and author responses from that submission.
Round 1
Reviewer 1 Report
Comments and Suggestions for Authors
The single center retrospective study compared the overall survival and complications in two cohorts of WHO high grade glioma patients with or without ventricle opening during the surgery. The main conclusion of the study that the ventricle opening is not associated with an increased risk of leptomeningeal dissemination, multicentric progression and hydrocephalus. The paper is well written, however there are some points that should be addressed before the paper is accepted.
1. The major concern of this study that the patient cohorts with or without ventricle opening were different and the total patient population was too heterogenous to make any conclusions. The statistical analysis needs to be improved. Probably mixed linear model should be used to analyze all factors.
2. The location of the tumor relative to the ventricles should be better classified. The MRI data of the most typical locations should be provided.
3. WHO grade III gliomas should be excluded from the study, or should be analyzed separately. MGMT methylation and IDH2 mutation status as well as other significant factors should be analyzed separately.
4. Line 104 – “The patients were divided into two groups according to whether or not the lateral ventricle was opened”. According this sentence, the ventricle opening was done in the first group.
5. Lines 189-190, The sentence is incorrect: «A survival analysis (Kaplan-Meier estimator) in patients with ventricular opening 189 versus those without opening at 6, 12 and 24 months showed a better OS». Ventricle opening showed lower OS and higher mortality risk. The significant difference in survival curves at 12 months should be discussed more carefully.
Author Response
The single center retrospective study compared the overall survival and complications in two cohorts of WHO high grade glioma patients with or without ventricle opening during the surgery. The main conclusion of the study that the ventricle opening is not associated with an increased risk of leptomeningeal dissemination, multicentric progression and hydrocephalus. The paper is well written, however there are some points that should be addressed before the paper is accepted.
We thank the reviewer for the feedback, and we will strive to address each point raised and modify the paper accordingly:
- The major concern of this study that the patient cohorts with or without ventricle opening were different and the total patient population was too heterogenous to make any conclusions. The statistical analysis needs to be improved. Probably mixed linear model should be used to analyze all factors.
Undoubtedly, implementing statistical analysis was necessary, which we have attempted to do in accordance with the reviewer's suggestions using a mixed linear model. However, I do not believe that the population is too heterogeneous in an absolute sense. In addressing the question "Is it more risky or advantageous to open the ventricle in high-grade gliomas?" we aimed to consider all patients who preoperatively appeared to have high-grade glioma lesions, rather than selectively choosing IDH wild-type glioblastomas post-surgery. This choice was made because the paper's objective is to provide preoperative guidance on whether to open the ventricle or not.
However, it is evident that the analysis needed to be stratified for various confounders, which we have done in this revised version of the paper. As can be seen from lines 190 to 210 and in Table 3, we have utilized a mixed linear model incorporating type and grade of neoplasm, as well as its distance from the ventricle, as confounders. The results, as expected, did not show correlations despite multiple stratifications
- The location of the tumor relative to the ventricles should be better classified. The MRI data of the most typical locations should be provided.
The information requested by the reviewer regarding the location and specific portion of the lateral ventricle affected by the neoplasm is provided in lines 144-152: "171 tumors were located in the right hemisphere (45%), 197 in the left (51.8%) and 12 along the midline (3.2%). Specifically, most lesions (162) were located precentrally in the frontal lobe (42.6%), 76 in the postcentral area (20%) and 142 in the temporo-insular region (37.4%). Considering the specific portion of the lateral ventricle next to tumor location, 34 (9.1%) lesions were adjacent to the occipital horn, 154 (41.1%) to the temporal horn, 124 (33.1%) to the frontal horn while the walls of the third ventricle were the nearest point in 63 cases (16.8%)." These details were obtained from preoperative magnetic resonance imaging (MRI).
Subsequently, in lines 153-156, the distance of the tumor from the ventricle is specified, regardless of the portion affected: "In 100 patients, the contrast-enhanced nodule was contiguous to the ventricle (26.3%); in the remaining patients, the distance was variable: <0.5 cm in 45 (11.8%), between 0.5 cm and 1 cm in 51 (13.4%), between 1 and 1.5 cm in 47 (12.4%), between 1 and 2 cm in 43 (11.3%), between 2 and 2.5 cm in 43 (11.3%) and over 2.5 cm in 51 patients (13.4%)." These distances, as specified in the Materials and Methods section, were also obtained from preoperative MRI.
- WHO grade III gliomas should be excluded from the study, or should be analyzed separately. MGMT methylation and IDH2 mutation status as well as other significant factors should be analyzed separately.
As specified in the response to point 1, we do not believe that tumors that were later revealed to be grade III on postoperative histological examination should be entirely excluded. This is because it is information that is not available before the surgical intervention. However, we fully agree that patients need to be stratified as this is a significant confounder. In response to this, we have added the following sentence at lines 192-196:
"A generalized mixed model was used to assess the potential association between ventricular opening and subsequent subependymal dissemination, considering confounding factors such as the type and grade of neoplasm, MGMT methylation profile, IDH mutation status, as well as its distance from the ventricle. The same analysis was then repeated using leptomeningeal dissemination, multicentric progression, and development of hydrocephalus or cerebrospinal fluid leak as outcomes."
- Line 104 – “The patients were divided into two groups according to whether or not the lateral ventricle was opened”. According this sentence, the ventricle opening was done in the first group.
We apologize for the oversight, we have revised the sentence as follows:
The patients were divided into two groups: one in which the lateral ventricle had not been opened and one in which it had been opened.
- Lines 189-190, The sentence is incorrect: «A survival analysis (Kaplan-Meier estimator) in patients with ventricular opening 189 versus those without opening at 6, 12 and 24 months showed a better OS». Ventricle opening showed lower OS and higher mortality risk. The significant difference in survival curves at 12 months should be discussed more carefully.
I believe there may be a typographical error that included the line number in the main text sentence, which now appears to be correct. In the discussion, we have modified lines 187 to 191 where we discussed the survival difference data, to emphasize how survival can be influenced by a myriad of factors (location, extent of resection, molecular factors, adjuvant therapy), and therefore, ventricular opening falls within this broad spectrum of variables. However, we expect it to influence survival in the event of complications rather than being a factor in itself.
Reviewer 2 Report
Comments and Suggestions for Authors
The authors present a manuscript regarding the impact of lateral ventricular opening in resection of HGG: this is a debated argument with a high potentiality but the design of this study is poor without scientific relevance.
The principle drawback of this paper is the lack of comparability among population: the authors subdivided the population in several categories with different tumor location, nearness to ventricle system and kind of ventricle opening; this choice reduces the scientific soundness limiting the translational relevance.
Other pitfalls should be explained by the authors:
- The inclusion criteria are arbitrary and must be supported by previous study or expert considerations.
- Pre and post op. tumor volume should be added in the Text; I don't agree the choice about the discrete steps among GTR and STR.
- The criteria about positioning LD or EVD should be discussed and explained, as well as the dural closure techniques.
- There is a discordance among the total amount of grade IV tumors and GBM or gliosarcomas. Similarly, the total amount of tumor adjacent to ventricle is incorrect.
- Please provide pre and post op. KPS and differentiate the PFS and OS for each subpopulation (sentences at 181/182 lines are not relevant).
- There is a contradiction between the first sentence among results at line 189-192 and subsequent ones.
Comments on the Quality of English LanguageA stylistic revision is advisable; please correct some grammatical mistakes.
Reviewer 3 Report
Comments and Suggestions for Authors
In this retrospective study report, the authors analyze the impact of opening of the ventricular system during the maximal surgery treatment of newly-diagnosed high-grade gliomas. The study included a relatively large cohort of patients (n=380), whose data were collected from a single center, and of which the opening of the lateral ventricles was required in 69 cases in order to maximize the removal of malignant tissue. As the authors state, this strategy was especially important to deploy in those cases with malignant tissue present inside or close to the subventricular zone (SVZ), a difficult area to manage surgically and a potential source of highly-infiltrative malignant glioma stem cells. In agreement with the previous literature reports, the authors did not find the opening of the ventricular system to be associated with a significant increase in post-operative complications (i.e., higher incidence of post-operative hydrocephalus or CSF leak, leptomeningeal and/or subependymal dissemination, or increased incidence of distal parenchymal recurrences) or to have a negative impact on the PFS. Importantly, the authors report that while the impact of lateral ventricle opening during surgery on the OS was trending downward compared to those patients who didn’t have the procedure, according to the included Kaplan-Meier survival data this negative trend did not reach statistical significance. While the authors acknowledge the limitations of their study (i.e., hard-to-generalize conclusions based on single-center data, lack of group randomization, CSF not routinely sampled and analyzed for cellularity, etc.), they conclude that the opening of the ventricular system is a procedure that is generally well-tolerated, advisable, and perfectly justified in those select cases that sanction a more aggressive approach to achieve maximal resection of malignant tissue. Overall, I found the article quite informative, concise, and well-written. I thank the authors for sharing their results.
My comment for the authors is whether they looked at the impact of chemoradiation (e.g., the Stupp protocol) on the post-operative complications following the opening of the ventricular system and/or the impact of overall OS (i.e., whether the non-surgical treatment played any role whatsoever). I am assuming that out of the 57 GBM patients who had surgery with ventricular opening, at least some of these patients received either the Stupp protocol (i.e., tumors with methylated MGMT promoters) or radiotherapy with or without CCNU (tumors with unmethylated MGMT promoters). In addition, I think a breakdown of the Kaplan-Meier data based on the post-operative treatment (i.e., TMZ-based chemoradiation vs. radiotherapy alone or with CCNU) would further benefit the potential reader of the study. I realize that the number of patients is probably not large enough to allow any statistically-meaningful conclusions to be drawn, but it would be interesting to see the trends (if any).
Minor comment: my understanding is that a number of 324 patients with histologically-confirmed GBM were included in the study. However, lines 134-135 state that “58 grade WHO grade III (15.3%) and 322 WHO grade IV (84.7%) gliomas were collected”. I am assuming that the number 322 is a typo that needs to be corrected.
Author Response
We thank the reviewer for the positive comments. Regarding the observation on adjuvant therapy, which we find interesting and relevant, we can make the following considerations:
Since all cases are primary diagnosis high-grade gliomas, CCNU-based regimens have never been used. Instead, only temozolomide-based regimens have been employed, with the choice between the Stupp regimen or the Perry regimen with hypofractionated radiotherapy being determined based on age, performance status, and extent of residual tumor (standard: 60 Gy in 30 doses; hypofractionated: 42 Gy in 14 doses).
We have added raw data on whether adjuvant therapy, both radiotherapy and chemotherapy, was administered or not at lines 169 and 171.
“A total of 281 (73,9%) patients received chemotherapy with Temozolomide (50 in the ventricle opening group), while 265 (69,7%) patients underwent postoperative radiotherapy (47 in the ventricle opening group), according to Stupp or Perry protocols.“
However, as expected, there was no statistically significant difference between ventricular opening or not and patients undergoing adjuvant treatments. This is likely because the administration of adjuvant therapy clearly affects survival, but we do not expect the intake of such drugs or the presence of radiotherapy to have an impact on complications. This is because radiotherapy is routinely initiated approximately 30 days after surgery, by which time the potential effects of ventricular opening on complications such as hydrocephalus or fistula have already occurred. As for dissemination, as the reviewer states, the number of patients with this kind of complications is too small to highlight a difference.
Minor comment: my understanding is that a number of 324 patients with histologically-confirmed GBM were included in the study. However, lines 134-135 state that “58 grade WHO grade III (15.3%) and 322 WHO grade IV (84.7%) gliomas were collected”. I am assuming that the number 322 is a typo that needs to be corrected.
We apologize for the error, it was a typing mistake indeed and we corrected it